# Research on Signal Feature Extraction of Natural Gas Pipeline Ball Valve Based on the NWTD-WP Algorithm

**DOI:** 10.3390/s23104790

**Published:** 2023-05-16

**Authors:** Lingxia Yang, Shuxun Li, Zhihui Wang, Jianjun Hou, Xuedong Zhang

**Affiliations:** 1School of Petrochemical Engineering, Lanzhou University of Technology, Lanzhou 730050, China; 202080706015@lut.edu.cn (Z.W.); 221080706001@lut.edu.cn (J.H.); 212080706009@lut.edu.cn (X.Z.); 2Machinery Industry Pump Special Valve Engineering Research Center, Lanzhou 730050, China

**Keywords:** natural gas, large-diameter pipeline ball valve, feature extraction, denoising, leakage signal

## Abstract

The measured signals of internal leakage detection of the large-diameter pipeline ball valve in natural gas pipeline systems usually contain background noise, which will affect the accuracy of internal leakage detection and sound localization of internal leakage points due to the interference of noise. Aiming at this problem, this paper proposes an NWTD-WP feature extraction algorithm by combining the wavelet packet (WP) algorithm and the improved two-parameter threshold quantization function. The results show that the WP algorithm has a good feature extraction effect on the valve leakage signal, and the improved threshold quantization function can avoid the defects of the traditional soft threshold function and hard threshold function, such as discontinuity and the pseudo-Gibbs phenomenon, when reconstructing the signal. The NWTD-WP algorithm is effective in extracting the features of the measured signals with low signal/noise ratio. The denoise effect is much better than that of the traditional soft and hard threshold quantization functions. It proved that the NWTD-WP algorithm can be used for studying the existing safety valve leakage vibration signals in the laboratory and the internal leakage signals of the scaled-down model of the large-diameter pipeline’s ball valve.

## 1. Introduction

Natural gas is the widespread energy source that is widely used for civil purposes due to its advantages in green environmental protection, economy, security, and reliability. Pipeline transportation has become the main means of transportation of natural gas by virtue of its advantages of low loss, safety, and reliability [1,2,3]. The high-parameter and large-diameter natural gas pipeline transportation system is constructed in order to improve the efficiency of natural gas pipeline transportation and reduce the costs of transportation. Therefore, the safety issues of air pollution and the potential explosion hazards caused by natural gas leakages in pipeline transportation systems are very important. The leakage of the natural gas pipeline system has caused serious economic losses and safety accidents [4]. According to reports, natural gas moves through a complex pipeline system, including elements such as pipes, valves, compression stations, and pressure regulation stations, etc., and the components that are more likely to leak are valves, flanges, and gaskets, as opposed to pipes [5,6,7]. As a key control element in long-distance pipelines, the pipeline ball valve plays an important role in the safe and stable operation of the pipeline system. If the valve is cut off and internal leakage occurs, it will cause a huge safety hazard. If the large-diameter pipeline ball valve can be accurately detected in time for internal leakage and the location of the internal leakage source can be accurately located and reported in time, the reliability and safety of natural gas pipeline transportation can be greatly improved, the risk of pipeline ball valve maintenance can be reduced, and the maintenance costs can be saved. The signal measured by the large-diameter pipeline ball valve internal leakage detection usually contains background noise, and the useful signal will be interfered with by noise, which affects the sound source localization accuracy of the internal leakage source. Therefore, it is particularly important to reasonably denoise the signal and extract its characteristic spectra to improve the accuracy of the subsequent signal analysis and reduce signal redundancy.

In order to improve the SNR and extract useful acoustic features, Ning F et al. decomposed the vibration signals of a Random Forest (RF) at different levels of leaks by Empirical Mode Decomposition (EMD) and Variational Modal Decomposition (VMD), respectively [8,9]. Wang Z et al. proposed a multi-source information fusion recognition method based on Variational Modal Decomposition (VMD) and Support Vector Machine (SVM) to solve the problem of the leakage of water pipes, and the VMD can decrease the collision noise [10,11]. It can be seen that EMD can adaptively decompose complex signals into a series of intrinsic mode functions; however, there is a problem of modal aliasing. VMD can effectively avoid the problems of EMD, but there are problems in the processing of boundary effects and burst signals. Goharrizi A Y applied a wavelet transform-based approach to detect internal leakage in a hydraulic actuator, and the results showed that the high-frequency part of the origin signal was not well-decomposed and analyzed [12]. In order to solve this problem, wavelet packet decomposition is proposed, which can effectively process all kinds of non-stationary random signals and can realize the feature decomposition of signals in different frequency bands at different times. With the strong ability of time–frequency localization decomposition of wavelet packet decomposition, it has been widely used in language, image, earthquake, mechanical vibration, and other fields. Yue G proposed the general Bayesian wavelet packet method based on the minimum mean square error and judged the denoising effect using a multi-factor analysis, and the results showed that the method has good denoising performance for weak information recognition [13]. Zhao X et al. performed feature extraction and denoising by analyzing the leakage acoustic emission signal with wavelet packets and predicted the internal leakage rate [14,15,16]. Pan Y proposed a structure health monitoring and assessment method based on the wavelet packet energy spectrum to diagnose the structure damage in time [17,18]. The research shows that the wavelet packet analysis method can effectively realize the reconstruction and feature extraction of the whole frequency band of the signal, but it has the shortcomings of insufficient self-adaptation and a relatively complicated calculation process. For this problem, Yu X developed a processing method of an ensemble empirical mode decomposition with adaptive noise (CEEMDAN) joint wavelet packet threshold to process ultrasonic, non-destructive testing defect signals, and concluded that the method can reduce the reconstruction error and iteration time and ensures the integrity of the signal. However, it has a relatively complex calculation process and has strict requirements on the installation position of the sensor [19].

In order to extract the useful information in the internal leakage signal and reduce the interference of signal redundancy and noise, this paper studies the feature extraction of internal leakage signals in the simulation of large-diameter pipeline ball valves. In which, the internal leakage signal refers to the vibration signal transmitted to the surface of the valve body by the elastic wave generated by the coupling between the internal leakage noise and the acoustic vibration of the valve and the valve control system, which is a secondary acoustic emission signal. The algorithms such as wavelet packet (WP) and wavelet threshold denoising (WTD) are studied, respectively. This paper proposes a new feature extraction algorithm that combines the improved two-parameter threshold quantization function and WP based on the WP and WTD algorithms. The measured valve internal leakage signal is denoised and extracted to obtain an accurate leakage signal.

## 2. Research Methodology

### 2.1. Wavelet Packet Feature Extraction Algorithm

#### 2.1.1. Principle

Wavelet packet decomposition can perform binary decompositions of low-frequency and high-frequency signals at the same time. High- and low-frequency signals are decomposed into low-frequency approximate signals and high-frequency detail information, respectively, so as to improve the resolution of the signal to be processed. Based on the wavelet packet’s single-branch reconstruction algorithm, when the number of decomposition layers is *n*, *n* ∈ Z, the frequency of *y*(*t*) is divided into 2*^n^* end node signals in turn. The length of the signal band of a node at the end is the ratio of *y*(*t*) band range to *n*, and the signal width of the corresponding sub-band of each layer is the same. When reconstructing the 2*^n^* end node coefficients of the *n*th layer of the wavelet packet decomposition structure tree, it is necessary to extract the signal within the frequency band of the corresponding single signal. *L_n_*_1_ represents the reconstructed signal of the end node coefficient low (*n*,1); *H_n_*_2_ is the reconstructed signal of the end node coefficient high (*n*,2); and the rest of the nodes are recursive. The reconstructed signal *y*(*t*) to be analyzed is shown in Equation (1), where *y*(*t*) is the original signal.
(1)$$y(t)=L_{n1}+H_{n2}+L_{n3}+\cdots \cdots +H_{2(2^{n}-1)}+L_{2(2^{n})}$$

WP relies on wavelet packet decomposition and single-branch reconstruction and has good signal resolution features in the time and frequency domains. Therefore, WP is used to extract the features of the internal leakage signal of large-diameter pipeline ball valves.

#### 2.1.2. WP Feature Extraction and Simulation

Common wavelet bases include Symlet series wavelets, Coiflet series wavelets, Daubechies series wavelets, Morlet series wavelets, etc. [20,21,22]. According to the investigation and analysis of the relevant literature, the wavelet bases commonly used in acoustic signal analysis include Db, Sym, Coif, and other series. DB series wavelet basis functions are selected for wavelet packet feature extraction [23].

The existing micro-open safety valve is used as the simulation signal to verify the feasibility of extracting the valve leakage signal using the WP algorithm. The vibration signal is excited by the internal leakage signal of the safety valve, which is transmitted to the surface of the valve body by the internal leakage hole and collected by the vibration sensor. WP feature extraction is carried out on the simulated signal to verify the feasibility of extracting the valve’s internal leakage signal using the WP feature extraction algorithm. Figure 1 shows the time domain and frequency domain diagram of the simulation signal.

It can be seen from Figure 1 that the internal leakage simulation signal of the safety valve only collects the internal leakage signal within 8000 Hz because the sampling frequency of the vibration acquisition instrument is 16,000 Hz. The signal frequency is mainly concentrated within 1000 Hz, and the feature extraction is mainly aimed at the 0~1000 Hz frequency band signal.

The Db6 wavelet basis function is selected for the WP analysis, and the number of decomposition layers is three. When reconstructing a single branch of the eight end node coefficients in the third layer of the wavelet decomposition structure tree, the signals in the frequency range of the corresponding single branch are extracted. *L*_30_ represents the reconstructed signal of the end node coefficients low (3,0); *H*_31_ is the reconstructed signal of the end node coefficients high (3,1); and so on for the rest of the nodes. The reconstructed signal to be analyzed, *y*(*t*), is shown in Equation (2).
(2)$$y(t)=L_{30}+H_{31}+L_{32}+H_{33}+L_{34}+H_{35}+L_{36}+H_{37}$$

In the formula, *y*(*t*) is the original signal; *L*_30_ is the reconstructed signal of terminal node 1 after a three-layer decomposition; *H*_31_ is the reconstructed signal of terminal node 2 after a three-layer decomposition; *L*_32_ is the reconstructed signal of terminal node 3 after a three-layer decomposition; *H*_33_ is the reconstructed signal of terminal node 4 after a three-layer decomposition; *L*_34_ is the reconstructed signal of terminal node 5 after a three-layer decomposition; *H*_35_ is the reconstructed signal of terminal node 6 after a three-layer decomposition; *L*_36_ is the reconstructed signal of terminal node 7 after a three-layer decomposition; and *H*_37_ is the reconstructed signal of terminal node 8 after a three-layer decomposition.

Based on the wavelet packet single-branch reconstruction algorithm, it can be seen that when the decomposition level is three, the frequency of *y*(*t*) is sequentially binarized and evenly distributed to the eight end node signals in the third layer. Assuming the frequency range of *y*(*t*) is [0, A], A ∈ C (C is a positive number), the diagram of the wavelet packet’s decomposition terminal node signal frequency band is shown in Figure 2.

The time domain diagram of each sub-band of the signal after WP decomposition is shown in Figure 3. As can be seen from Figure 3, the simulation signal is decomposed into eight sub-bands of equal frequency band through the three-layer WP decomposition. Secondly, the energy value is extracted through the wavelet coefficient of the end node as the feature of the signal [24]. The energy of the end node is usually expressed as the percentage of the energy value of each node in the total energy, so as to construct the wavelet packet energy spectrum of the end node, and select the sub-band with the largest energy spectrum as the characteristic spectra of the simulation signal to extract its feature. The time domain diagram of the end node after WP decomposition and reconstruction is shown in Figure 3. Through the time domain diagram of the signals in the eight sub-bands decomposed by the three-layer WP, it can be seen that the distribution of the waveform voltage signals in the last seven sub-bands is similar, with higher amplitudes in the 23 s and 29–32 s intervals, especially in the fifth sub-band. The maximum amplitude of the first four sub-bands approaches 20 mV, the maximum amplitude of the fifth, sixth, and seventh sub-bands approaches 10 mV, and the maximum amplitude of the eighth sub-band is around 5 mV. The overall fluctuation of the waveform voltage signal distribution in the first sub-band is not significant.

It can be seen from Figure 4 that the energy of the first sub-band accounts for 70.42%, and the energy accounts for the largest proportion among the eight sub-bands. Therefore, the first sub-band is selected as the feature extraction band to be reconstructed, and the time domain and frequency domain waveforms of the characteristic spectra are shown in Figure 5. It can be seen from Figure 5 that after WP extracts the feature band, only the first sub-band signal with the largest energy proportion is retained, which has a good feature extraction effect. It is verified that the WP feature extraction method can be used for the analysis of valve leakage signals.

### 2.2. The Improved Wavelet Threshold Denoising

#### 2.2.1. Wavelet Threshold Denoising

The purpose of WTD is to eliminate the noise component in the signal *y*(*t*) to be analyzed as much as possible and retain the useful acoustic emission signal *v*. The bottom model of *y*(*t*) is shown in Equation (3).
(3)y(t)=f(t)+σe(t)
where *f*(*t*) is the useful acoustic signal; *e*(*t*) is the noise signal; and *σ* represents the level of noise.

The principle of WTD is to wavelet transform the signal *y*(*t*) to be analyzed. If the absolute value of the wavelet coefficient is large, the proportion of useful signals in the wavelet coefficient is greater than that of noise signals. Otherwise, the noise signal is dominant. Therefore, the threshold *λ* can be set according to the value of the wavelet coefficient, and the wavelet coefficients that are smaller than λ can be eliminated to realize the noise reduction function. The steps of WTD mainly include the following:(1)The appropriate decomposition level *n* is selected according to the time and frequency feature. The signal wavelet coefficients are obtained from wavelet transform *y*(*t*). Unlike from wavelet packet decomposition, wavelet decomposition only performs binary decomposition for low-frequency approximate signals at each scale.(2)Select a reasonable threshold function and threshold quantization function to filter the wavelet coefficients of high-frequency detail signals under different decomposition scales, and retain the low-frequency approximate signals under the highest scale.(3)The filtered wavelet coefficients are transformed by inverse wavelet transform to obtain the reconstructed signal after threshold denoising. The mathematical model of the reconstructed signal *y**(*t*) is shown in Equation (4).
(4)$$y^{*}(t)=L_{3}+H_{3}+H_{2}+H_{1}$$
where *y**(*t*) is the reconstructed signal after reducing noise and reconstructing the signal *y*(*t*); *L*_3_ is the low-frequency approximate signal after a three-layer decomposition. *H*_3_ is the high-frequency detail signal after a three-layer decomposition; *H*_2_ is the high-frequency detail signal after a two-layer decomposition; and *H*_1_ is the high-frequency detail signal after one layer decomposition.

#### 2.2.2. Improved Adaptive Threshold Function

The value of threshold *λ* determines the effect of WTD. If the value of *λ* is too large, the useful acoustic emission signal will be judged as noise signal and filtered out. If the value of *λ* is too small, it will result in incomplete noise reduction. The commonly used threshold algorithms mainly include the Sqtwlolg threshold function, Rigrsure threshold function, Minimax threshold function, and Heursure threshold function [25,26,27,28].

After the wavelet transform of *y*(*t*), the noise energy of different decomposition scales accounts for different proportions in the wavelet coefficients of the corresponding scales. In order to improve the effect of signal noise reduction, different values of threshold *λ* should be obtained for the noise energy at different scales. Since the value of threshold *λ* can be obtained adaptively by the Rigrsure threshold function through wavelet coefficients at different scales. Therefore, the threshold function is selected for a noise reduction analysis.

The Rigrsure threshold function is an adaptive threshold acquisition algorithm. Firstly, it makes the absolute value of each element *x_i_* of the signal and arranges them in ascending order. Secondly, it squares each element to obtain a new sequence *r*(*x_i_*), as shown in Equation (5).
(5)$$r(x_{i})={\left(sort(\left|x_{i}\right|)\right)}^{2}$$
where *i* = 1, 2, 3, …, *N*; |*x*_1_| ≤ |*x*_2_| ≤⋯⋯≤ |*x*_N_|.

If *λ* is the square root of the *i*th element of the new sequence *r*(*x_i_*), then the expression of the Rigrsure threshold function is as shown in Equation (6).
(6)$$\lambda_{r}=\zeta \sqrt{r(x_{i})}$$

Rish(*i*) is called the threshold acquisition risk, and its mathematical expression is shown in Equation (7).
(7)$$Rish(i)=\frac{N-2i+\sum r(x_{i})+(N-x)r(x_{i})}{N}$$

Combining Equations (5) to (7), it can be known that the value of threshold *λ* can be obtained adaptively by the Rigrsure threshold function through the wavelet coefficients under different scales. However, this function is not sensitive to the general law that the noise decreases with increases in the decomposition scale, and a reasonable improvement is necessary for this problem.

The underlying mathematical model of the *y*(*t*) signal in Equation (1) is regarded as the M-dimensional random vector model, as shown in Equation (8).
(8)$$\left[\begin{array}{c} Y(0)\\ Y(1)\\ Y(2)\\ \vdots \\ Y(M-1) \end{array}\right]=\left[\begin{array}{c} F(0)\\ F(1)\\ F(2)\\ \vdots \\ F(M-1) \end{array}\right]+\left[\begin{array}{c} \sigma E(0)\\ \sigma E(1)\\ \sigma E(2)\\ \vdots \\ \sigma E(M-1) \end{array}\right]$$

Let the coefficients of the matrix vector of Equation (8) after WT be three sets of vectors ***d****^*^_j,k_*, ***f****_j,k_*, and ***e***, respectively, and the corresponding mathematical model is shown in Equation (9).
(9)$$d^{*}{{}_{j,}}_{x_{i}}=f_{j,x_{i}}+e$$
where *j* ∈ *Z*, which represents different decomposition scales.

Substitute the wavelet coefficients at different scales obtained in Equation (9) into Equation (7) to obtain the risk threshold vector Rish(*i*), and then Rish(*i*) min is obtained according to the min algorithm, which is substituted into Equation (5) to obtain the Rigrsure threshold *λ_j_* under different decomposition scales. The thresholds on different scales are as shown in Equation (10).
(10)$$\lambda_{j}=\delta_{j}\sqrt{r{\left(x_{i}\right)}_{min}}$$
where *δ_j_* is the standard deviation of noise at different scales.

The proportion of noise decreases with increases in the decomposition scale. If the calculated value of threshold *λ* is large, some useful acoustic emission signal coefficients will be filtered out, resulting in signal distortion. The Rigrsure threshold function is improved to obtain different values of threshold *λ* according to different decomposition scales by referring to the method of reference [29], and it is calculated as Equation (11).
(11)$$\lambda_{j}=\frac{\delta_{j}\sqrt{r{\left(x_{i}\right)}_{\min}}}{\log(j+1)}$$

It can be seen from Equation (11) that the changes in the new vector *r*(*x_i_*) and decomposition scale *j* are recorded according to the wavelet coefficients of each scale, which is improved based on the Rigrsure threshold function of Equation (6). The optimal *λ* is obtained adaptively at each scale, and the signal is denoised more reasonably.

#### 2.2.3. Improved Two-Parameter Threshold Denoising

The wavelet coefficients need to be analyzed using a threshold quantization function when the threshold *λ* is obtained. The hard threshold function only retains wavelet coefficients greater than *λ* and sets the rest of the wavelet coefficients to zero. The mathematical expression is expressed in Equation (12).
(12)$$y_{h}(d,\lambda )=\left\{\begin{array}{c} d\\ 0 \end{array}\right.\begin{array}{ll} & \left(\left|d\right|>\lambda \right)\\ & \left(\left|d\right|\leq \lambda \right) \end{array}$$
where *λ* is the threshold at the scale *j*; *d* is the wavelet coefficient of the signal after WT.

The soft threshold function sets wavelet coefficients that are smaller than *λ* to zero and shrinks the rest of the wavelet coefficients toward zero. The mathematical expression is shown in Equation (13).
(13)$$y_{s}(d,\lambda )=\left\{\begin{array}{ll} \mathrm{sign}(d)(\left|d-\lambda \right|) & \left(\left|d\right|>\lambda \right)\\ 0 & \left(|d|\leq a\right) \end{array}\right.$$

The edge of the signal is well preserved in the traditional hard threshold quantization function analysis. However, there are two discontinuous points in the wavelet domain, and the estimated wavelet coefficients obtained are discontinuous, which would cause signal distortion and an unsatisfactory noise reduction effect. The soft threshold quantization function has a continuous structure, and the processed signal is smooth, but not differentiable at *λ*, and has a constant deviation from the true wavelet coefficients.

An adaptive threshold quantization function with a two-parameter adjustment factor is proposed to filter out complex background noise in acoustic signals, in view of the defects of discontinuity and the inherent bias of soft and hard threshold quantization functions. The improved threshold quantization function can reasonably analyze wavelet coefficients that are larger than *λ* by adjusting the two parameters. While ensuring the continuity of the function, avoid directly setting the wavelet coefficients as smaller than *λ* to zero, which prevents the phenomenon of oscillation generating pseudo-Gibbs during signal reconstruction. The improved threshold quantization function is shown in Equation (14).
(14)$$d^{*}{}_{j,k}(d_{j,k},\alpha ,\beta )=\left\{\begin{array}{cc} \mathrm{sign}(d_{j,k})(\left|d_{j,k}\right|-\frac{e^{\alpha -1}}{e^{\beta }}\lambda ), & \left|d_{j,k}\right|\geq \lambda \\ \mathrm{sign}(d_{j,k})(e^{\alpha (\left|d_{j,k}\right|-\lambda )}\times \left|d_{j,k}\right|-\frac{e^{\alpha -1}}{e^{\beta }}\lambda ), & \left|d_{j,k}\right|<\lambda \end{array}\right.$$
where sign (*) is a signum function; *d_j_*_,*k*_ is the original wavelet coefficient; *d^*^_j_*_,*k*_ is the wavelet coefficient after noise reduction; *λ* is the wavelet threshold; and *α* and *β* are the two-parameter adjustment factors, where 0 < *α* ≤ 1, *β* ∈ N*||*β* = 0, and *β* = 100*α*.

It can be seen from Equation (14) that the wavelet coefficient of the true signal is larger when |*d_j,k_*| > *λ*, and the noise component accounts for a smaller proportion than the true wavelet coefficient. In order to improve the noise reduction effect, the two-parameter adjustment factors α and β are introduced to fine-tune the noise signal when |*d_j,k_*| > *λ*. When *α* → 0, *β* → +∞, the improved threshold quantization function is biased toward the hard threshold. When *α* = 1, *β* = 0, the improved threshold quantization function degenerates to a soft threshold quantization function.

The useful wavelet coefficients can be retained to the maximum extent when |*d_j_*_,*k*_*|* < *λ* and the wavelet threshold *λ* is close to the wavelet coefficients *d_j_*_,*k*_. Noise is dominant with the reduction in wavelet coefficients. This part of the information is eliminated by using the fast attenuation characteristic of the exponential function. In addition, through the two parameters *α* and *β*, the improved threshold quantization function can be adjusted based on soft and hard thresholds according to different signal feature, which improves the adaptability of the threshold quantization function.

The simulation studies show that it is suitable for analyzing signals with low noise interference and a high signal-to-noise ratio when the parameters *α* and *β* of the improved threshold quantization function are assigned small values, which is conducive to maintaining the detail component of the signal. For signals with more noise components and a low signal-to-noise ratio, a smaller value of *α* and a larger value of *β* are generally beneficial to filter out more noise components. Additionally, when the two parameters satisfy *β* = 100*α*, the improved threshold quantization function has the best continuity and a smoother curve.

The comparison chart of the improved threshold quantization function curve and the soft and hard threshold quantization function curves are shown in Figure 6, when the two-parameter adjustment factors of the improved threshold quantization function are *α* = 0.03, *β* = 3, *α* = 0.1, *β* = 10, *α* = 0.25, *β* = 25, *α* = 0.5, and *β* = 50. It can be seen from Figure 6 that when adjusting the two parameters of the improved threshold quantization function, the threshold quantization function curve is located between the soft and hard threshold quantization function curves and is closer to the hard threshold quantization function curve. When the original wavelet coefficients are within the interval [−15,15], the function curve tends to be flat and smooth, and it tends to a linear relationship with a constant slope when the adjustment factor *α* → 0 and *d_j,k_* ∈ (−*λ*, *λ*); when α → 1 and *d_j,k_* ∈ (−*λ*, *λ*), the function curve tends to be steep, and its value is closer to the hard threshold quantization function and the soft threshold function. When *α* → 0 and *β* → +∞, the quantitative relationship between *α* and *β* satisfies *β* = 100*α*; that is, the improved function curve of the value of *α* = 0.03, *β* = 3 with the best continuity and a smoother curve.

The continuity of the improved threshold quantization function is verified, and *d_j,k_* = *λ* and *d_j,k_* = −*λ* at coordinates are verified function continuity, respectively.

(1) When *d_j_*_,*k*_ = *λ*:

① when *d_j,k_*→*λ*^+^, 0 < *α* < 1, and *β* ∈ N^*^ meet *β* = 100*α*, the left part of function is as follows:(15)$$d^{*}{}_{j,k}(\lambda^{+},\alpha ,\beta )=\mathrm{sign}(\lambda^{+})(\left|\lambda^{+}\right|-\frac{e^{\alpha -1}}{e^{\beta }}\lambda )=\lambda^{+}-\frac{e^{\alpha -1}}{e^{\beta }}\lambda \approx \lambda$$

② when *d_j,k_*→*λ*^−^, 0 < *α* < 1, and *β* ∈ N^*^ meet *β* = 100*α*, the right part of function is as follows:(16)$$d^{*}{}_{j,k}(\lambda^{-},\alpha ,\beta )=\mathrm{sign}(\lambda^{-})e^{\alpha (\left|\chi \right|-\lambda )}\times \left|\lambda^{-}\right|-\frac{e^{\alpha -1}}{e^{\beta }}\lambda \approx \lambda$$

③ when *d_j,k_* → *λ*^+^, *α* = 1, and *β* = 0, the left part of function is as follows:(17)$$d^{*}{}_{j,k}(\lambda^{+},\alpha ,\beta )=\mathrm{sign}(\lambda^{+})(\left|\lambda^{+}\right|-\frac{e^{1-1}}{e^{0}})\lambda =\lambda^{+}-\lambda =0$$

④ when *d_j,k_*→*λ*^−^, *α* = 1, and *β* = 0, the right part of function is as follows:(18)$$d^{*}{}_{j,k}(\lambda^{-},\alpha ,\beta )=\mathrm{sign}(\lambda^{-})\cdot e^{\left(\left|\chi \right|-\lambda \right)}\times \left|\lambda^{-}\right|-\frac{e^{1-1}}{e^{0}}\lambda =0$$

According to Equations (14)–(17), when *d_j_*_,*k*_ = *λ*, within the range of *α* ∈ (0,1], *β* ∈ N*|| *β* = 0, Equation (14) is equal to Equation (15), and Equation (16) is equal to Equation (17); so, in *d_j_*_,*k*_ = *λ*, the improved threshold quantization function is continuous.

(2) When *d_j,k_* = −*λ*:

① When *d_j,k_* → (−*λ*) ^+^, 0 < *α* < 1, *β* ∈ N^*^, and *β* = 100*α*, the left part of function is as follows:(19)$$d^{*}{}_{j,k}({\left(-\lambda \right)}^{+},\alpha ,\beta )=\mathrm{sign}({\left(-\lambda \right)}^{+})(\left|{\left(-\lambda \right)}^{+}\right|-\frac{e^{\alpha -1}}{e^{\beta }}\lambda )\approx -(\lambda^{+}-0\times \lambda )\approx -\lambda$$

② When *d_j,k_* → (−*λ*)^−^, 0 < *α* < 1, *β* ∈ N*, and *β* = 100α, the right part of function is as follows:(20)$$d^{*}{}_{j,k}({\left(-\lambda \right)}^{-},\alpha ,\beta )=\mathrm{sign}({\left(-\lambda \right)}^{-})\cdot e^{\alpha (\left|{\left(-\lambda \right)}^{-}\right|-\lambda )}\times \left|{\left(-\lambda \right)}^{-}\right|-\frac{e^{\alpha -1}}{e^{\beta }}\lambda =-(\lambda^{-}-0\times \lambda )\approx -\lambda$$

③ When *d_j,k_* → (−*λ*)^+^, *α* = 1, and *β* = 0, the left part of function is as follows:(21)$$d^{*}{}_{j,k}({\left(-\lambda \right)}^{+},\alpha ,\beta )=\mathrm{sign}({\left(-\lambda \right)}^{+})(\left|{\left(-\lambda \right)}^{-}\right|-\frac{e^{1-1}}{e^{0}}\lambda )=-(\lambda^{+}-\lambda )=0$$

④ When *d_j,k_* → (−*λ*)^−^, *α* = 1, and *β* = 0, the right part of function is as follows:(22)$$d^{*}{}_{j,k}({\left(-\lambda \right)}^{-},\alpha ,\beta )=\mathrm{sign}({\left(-\lambda \right)}^{-})\cdot e^{\left(\left|{\left(-\lambda \right)}^{-}\right|-\lambda \right)}\times \left|{\left(-\lambda \right)}^{-}\right|-\frac{e^{1-1}}{e^{0}}\lambda =-(\lambda^{-}-\lambda )=0$$

According to Equations (19)–(22), when *d_j_*_,*k*_ = −*λ*, within the range of *α* ∈ (0,1], *β* ∈ N*|| *β* = 0, Equation (19) is equal to Equation (20), and Equation (21) is equal to Equation (22); so, in *d_j_*_,*k*_ = *λ*, the improved threshold quantization function is continuous.

To sum up, the improved threshold quantization functions in *d_j_*_,*k*_ = *λ* and *d_j_*_,*k*_ = −*λ* are continuous. Therefore, the improved threshold quantization function is continuous in the whole positive and negative wavelet domain. It avoids the discontinuity of traditional soft and hard threshold quantization functions and the Pseudo Gibbs phenomenon in signal reconstruction.

In order to verify the effectiveness of the noise reductions in the improved threshold quantization function, the simulated signal in Section 2.1.2 (the signal contains noise) is also used to denoise the signal by the traditional soft threshold, hard threshold, and improved threshold quantization functions, respectively. The number of decomposition layers in the simulation experiment is selected by trial calculations. The improved Rigrsure threshold function is selected and the expression of the function is shown in Equation (11). The Db6 series is selected to be the wavelet basis function. The signal after noise reduction in the simulated signal by three different threshold quantization functions is shown in Figure 7.

It can be seen from Figure 7 that the noise reduction in the hard threshold function is seriously distorted due to the discontinuity of the threshold function, especially in the range of *t* ∈ [24.38, 28.95], the signal amplitude is quite different from the raw signal amplitude. Taking the extreme values of *t* as examples, when *t* = 24.38 s, the amplitude difference from the raw signal is −0.29 after using the hard threshold function for noise reduction, and the amplitude difference at *t* = 28.95 s is 0.17. The signal amplitude of the soft threshold function has a certain deviation from the raw signal in the time domain range. When *t* = 24.38 s, the difference from the raw signal amplitude is 3.25, and when *t* = 28.95 s, the difference from the raw signal amplitude is 3.35. The signal distortion is the smallest after using the improved threshold quantization function for noise reduction. When *t* = 24.38 s, the difference from the raw signal is −0.007, and when *t* = 28.95 s, the difference between the amplitude of the signal and the raw signal is 0.007. The signal amplitude is close to the raw signal, avoiding the defect of excessive noise reduction in the traditional soft and hard threshold.

In order to quantitatively compare and analyze the advantages and disadvantages of different threshold functions for noise reduction, Signal Noise Ratio (SNR) and Root Mean Squared Error (RMSE) are used as the evaluation criteria for noise reduction. Equation (23) and Equation (24) are the mathematical expressions of SNR and RMSE, respectively.
(23)$$\mathrm{SNR}=10\mathrm{lg}\left[\frac{\sum_{i=1}^{n}x^{2}(t)}{\frac{\sum_{i=1}^{n}{\left[x(t)-d(t)\right]}^{2}}{L}}\right]$$
(24)$$\mathrm{RMSE}=\sqrt{\frac{\sum_{i=1}^{n}{\left[x(t)-d(t)\right]}^{2}}{L}}$$
where *x*(*t*) is the raw signal; *d*(*t*) is the signal after noise reduction; and *L* is the length of signal.

Table 1 shows a comparison of the noise reduction effect of different threshold quantization functions.

As can be seen from Table 1, when the two-parameter adjustment factors *α* and *β* of the improved threshold quantization function satisfy *β* = 100*α*, the value of SNR after noise reduction in the improved threshold quantization function is 2.35 times, 2.24 times, 1.81 times and 1.54 times of the traditional soft threshold function, respectively, and 3.26 times, 3.11 times, 2.51 times and 2.13 times of the hard threshold function, respectively. The value of RMSE after noise reduction in the improved threshold quantization function is 11.08 times, 9.16 times, 4.40 times and 2.58 times smaller than the traditional soft threshold function, respectively, and 17.82 times, 14.75 times, 6.84 times and 4.21 times smaller than the hard threshold function, respectively. Therefore, the noise reduction effect of the improved threshold quantization function is much better than that of the traditional soft and hard thresholding functions. Additionally, when *α* → 0, the improved threshold noise reduction effect is the best. In addition, when *α* = 1 and *β* = 0, the noise reduction effect of the improved threshold quantization function is better than that of the hard threshold and worse than that of the soft threshold. After a large number of simulation experiments, it is determined that a set of two parameters with the best noise reduction effect, namely *α* = 0.03, *β* = 3, are used as the dual parameter values of the improved threshold quantization function. In summary, through the comparative analysis of the noise reduction in different threshold quantization functions, the feasibility of the noise reduction in the improved threshold quantization function is verified, and the noise reduction effect is much better than the traditional soft and hard threshold functions.

### 2.3. The Feature Extraction Algorithm

In order to extract useful information from the internal leakage signal and reduce the interference of signal redundancy and noise, a feature extraction algorithm (NWTD-WP) based on an improved wavelet threshold denoising function combined with the wavelet packet is proposed. It performs wavelet threshold noise reduction on the signal to filter out the background noise in the signal and selects the improved two-parameter threshold quantization function to bring the denoised signal into WP for feature band extraction.

The simulated signal in Section 2.1.2 is taken as the analysis object to verify the feasibility of the NWTD-WP algorithm to extract signal feature information. The signal after noise reduction by the improved two-parameter threshold quantization function (two-parameter *α* = 0.03, *β* = 3) is taken as the input and brought into the WP algorithm, and the Db6 wavelet basis is selected with three decomposition layers. Figure 8 shows the energy spectrum of the end node signal after WP decomposition. It can be seen from Figure 7 that when feature extraction is performed after noise reduction in the improved threshold function, the energy of signal sub-band one accounts for 73.19%, which is 2.77% higher than that of WP feature extraction. Therefore, the NWTD-WP feature extraction algorithm has a better feature extraction effect for the actual measured internal leakage signal of the valve; that is, when the feature extraction is performed on the signal containing the noise component itself.

Figure 9 depicts the time domain and frequency domain diagram of the first sub-band after reconstruction. It can be seen from Figure 9 that the simulated signal after NWTD-WP feature extraction only retains the main frequencies before 1000 Hz, which has a good feature extraction effect and verifies the feasibility of the NWTD-WP feature extraction algorithm.

## 3. Experiment

### 3.1. Experimental Setting

Based on the principle of geometrical similarity, the components of the large-diameter pipeline ball valve, such as the middle valve body, left valve body, seat, and seat seal, which have a large impact on the propagation of the internal leakage signal, are designed in a 1:3 scale. Figure 10 depicts a schematic diagram of the internal leakage signal acquisition system for the scaled-down model of a large-diameter pipeline ball valve.

As can be seen from Figure 10, the acquisition system mainly consists of a scaled-down model of a large-diameter pipeline ball valve, four acoustic emission sensors, a multi-channel, high-frequency acoustic emission instrument, four pre-signal amplifiers, and a PC terminal. The nominal diameter of the scaled model is DN200, the vertical distance of the UCA array *XOY* section from the internal leakage source is 50 mm, the array radius R = 110 mm, and the centra *O* of the coordinate system is located on the axis of the circular array. Let the angle between the line of the internal leakage source and the origin *O* (dashed line in the diagram) and the *X*-axis be the azimuth angle *θ* and the angle with the *Z*-axis be the pitch angle *φ*.

Figure 11 is a multi-channel, high-frequency dynamic acoustic emission instrument with a sampling frequency of up to 3 MHz, which enables real-time full waveform acquisitions of acoustic emission signals and observations of the panoramic contours and waveform details of the acquired signals. Parameters of high-frequency acoustic emission sensor is shown in Table 2. The signal amplifier shown in Figure 11 is mainly used to amplify the signal collected by the acoustic emission sensor to prevent signal attenuation. A pre-amplifier with a gain of 40 dB is used to gain the internal leakage signal collected by the sensor. The acoustic emission sensor model shown in Figure 12 is RS-13A, with a sampling frequency range of 16 kHz to 60 kHz and a sensor center frequency of 40 kHz. The right image in Figure 12 is a high vacuum coupling silicone grease, using this coupling agent can fill the small gap between the sensor contact surfaces, playing a transitional role and reducing the signal attenuation loss.

### 3.2. Signal Acquisition and Feature Extraction

The incident sources of the internal leakage signal of the reduced scale model of the large-diameter pipeline ball valve are preset. Based on the signal acquisition system, the lead breaking signal at the sealing ring of the valve seat of the reduced scale model under the corresponding incident source is collected, respectively, to replace the measured internal leakage signal of the valve. In order to reduce the experimental error, the lead breaking angle of the valve seat sealing ring of the reduced scale model is selected as 30° according to the standard. The time domain and frequency domain waveforms of the measured signals collected by the four acoustic emission sensors are shown in Figure 13.

As can be seen from Figure 13, at the moment of medium leakage due to a structure break, the signals received from acoustic emission Sensor 1 to Sensor 4 with a concentration of energy around 0.07 s in the time domain distribution. The peak energy of Sensor 1 and Sensor 3 is about 7 mV, and the peak energy of Sensor 2, which is closest to the internal leakage source, is the highest at 8.45 mV. In the frequency domain distribution of the received signal, peaks appear at frequencies such as 8.8 kHz, 15 kHz, 19 kHz, and 49 kHz, and the signal energy amplitude is relatively high. The signals received by the four sensors have relatively high energy amplitudes in the frequency domain distribution, ranging from 45 kHz to 52 kHz. Since the measured signal contains background noise, the frequency band before 20 kHz is not extracted as the signal characteristic spectra to avoid the influence of noise on the positioning accuracy. The NWTD-WP feature extraction algorithm is used to extract the high-frequency feature signal near 49 kHz as the data input for the positioning of the measured signal.

The Db6 wavelet basis function is selected to extract signal features, and the number of wavelet packet decomposition layers is three. The improved threshold quantization function is selected, where *α* is 0.03 and *β* is 3, and the improved threshold quantization function is used to obtain the threshold value. Keeping the same feature extraction method as in Section 2.3, and extracting the feature signal in the range of 45 kHz~52.5 kHz in the seventh sub-band. The time and frequency domains of the seventh sub-band signal after the NWTD-WP feature extraction of the received signals of the four sensors are shown in Figure 14.

From the time domain waveforms in Figure 14a–d, the signal after feature extraction retains the main components of the signal, and the waveform is consistent with the original waveform. The frequency of the signal after feature extraction is concentrated around 45 kHz~52.5 kHz, and the attenuation of the energy amplitude is small. Therefore, the NWTD-WP feature extraction algorithm is used to denoise the measured signals collected by the four sensors with an improved two-parameter threshold function, and then combined with the wavelet packet feature extraction algorithm, the feature signals of the seventh sub-band from 45 kHz to 52.5 kHz are successfully extracted.

## 4. Conclusions

Combined with the WP algorithm and the improved two-parameter threshold quantization denoising function, an NWTD-WP feature extraction algorithm is proposed. The existing vibration signal of the safety valve and the signal measured by the internal leakage test of the scaled-down model of the large-diameter pipeline ball valve are, respectively, studied for WP and NWTD-WP feature extraction. The results show that the established improved threshold quantization function is continuous in the whole wavelet domain. When the two parameters of the improved threshold quantization function are *α* = 0.03 and *β* = 3, the denoising effect is optimal. When the improved threshold quantization function is used for denoising, the signal distortion after denoising is the smallest, and the signal amplitude is close to the raw signal, which avoids the defects of the traditional soft and hard threshold functions, such as discontinuity and the Pseudo-Gibbs phenomenon, when reconstructing the signal. The denoise effect is much better than that of the traditional soft and hard threshold quantization functions. For the measured signal with a low signal/noise ratio, the NWTD-WP algorithm can effectively extract the feature signal of the sub-band, and the feature extraction effect is better. However, the limitation of the NWTD-WP algorithm is that the effectiveness of the feature extraction in NWTD-WP depends on the value of the two parameters of the improved threshold quantization function. The proposed NWTD-WP algorithm can be extended to feature extraction and denoising mechanical equipment leakage signals in industrial pipeline valve systems.

## Figures and Tables

**Figure 1 sensors-23-04790-f001:**
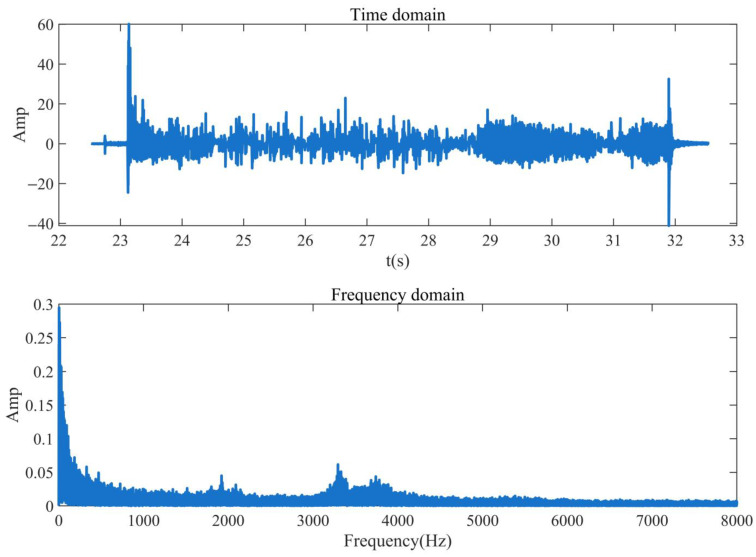
Time domain and frequency domain waveforms of signal.

**Figure 2 sensors-23-04790-f002:**
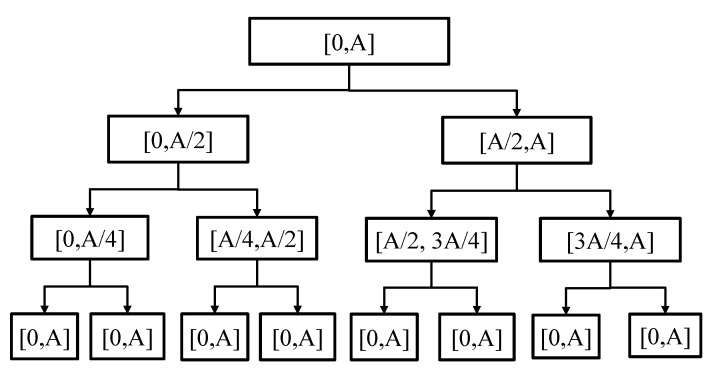
Frequency band distribution of terminal node signal in wavelet packet decomposition.

**Figure 3 sensors-23-04790-f003:**
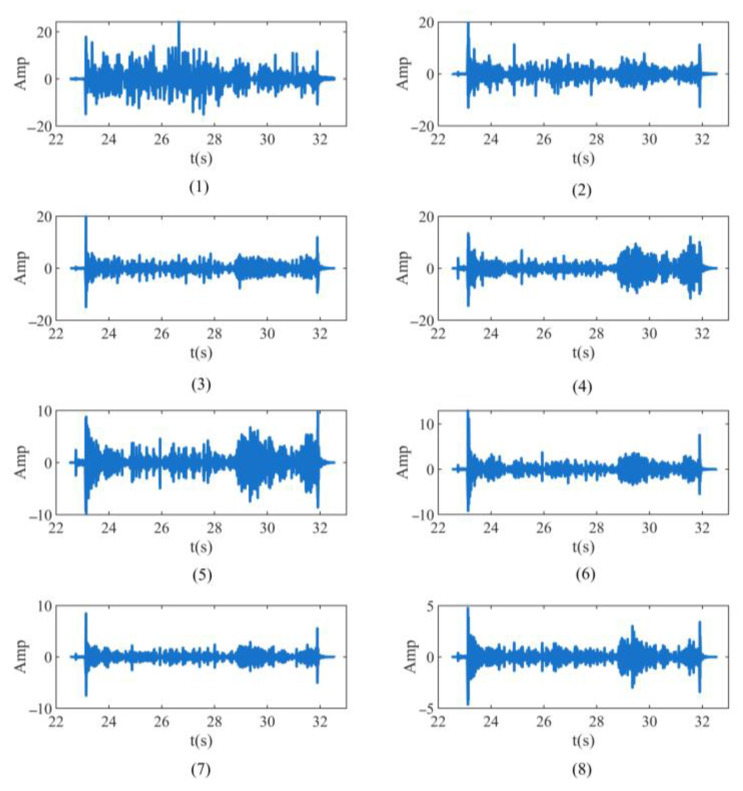
Time domain waveforms of each sub-band after three-layer wavelet decomposition of signal.

**Figure 4 sensors-23-04790-f004:**
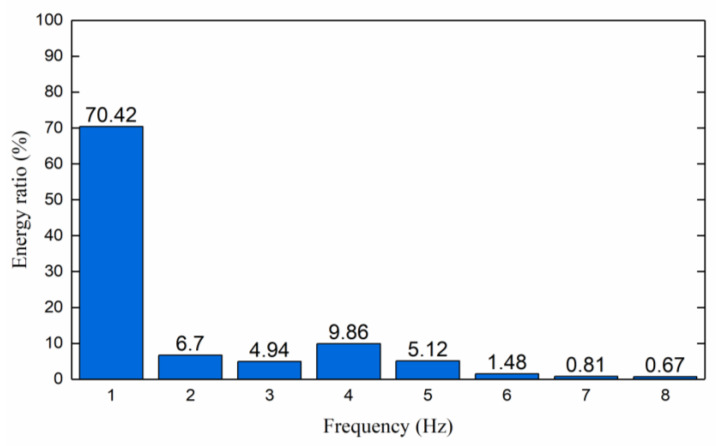
The energy ratio of each sub-band after wavelet decomposition of the signal.

**Figure 5 sensors-23-04790-f005:**
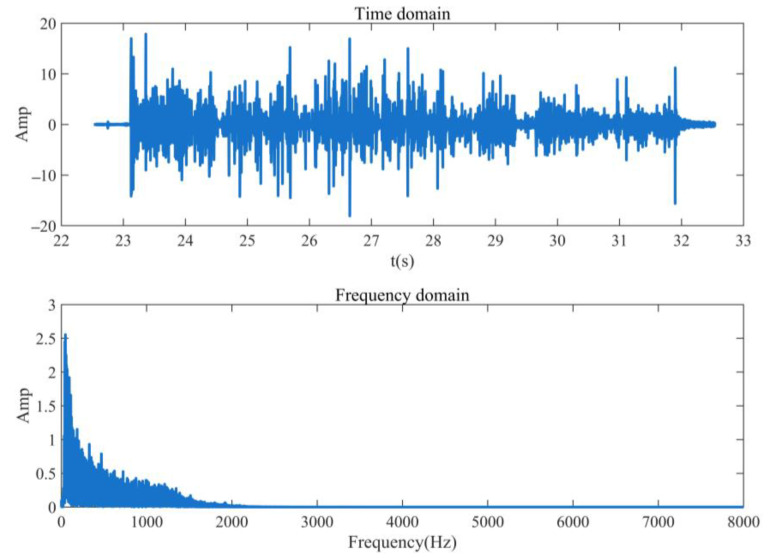
Time domain and frequency domain waveforms of the first sub-band signal in WP feature extraction.

**Figure 6 sensors-23-04790-f006:**
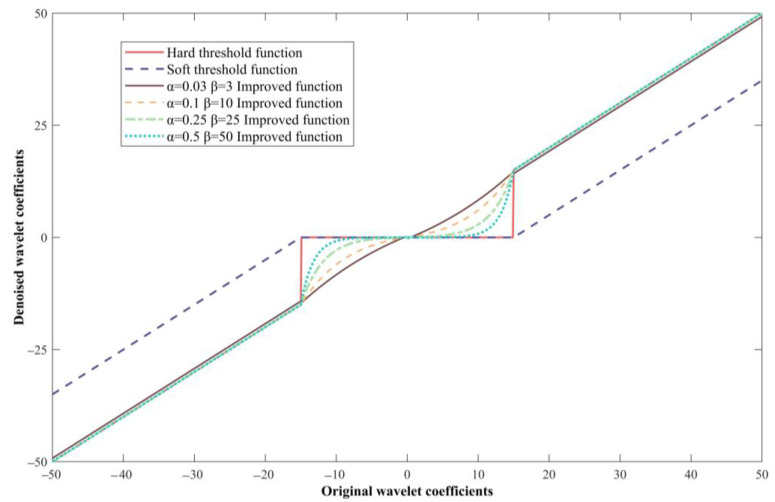
Threshold function curve comparison.

**Figure 7 sensors-23-04790-f007:**
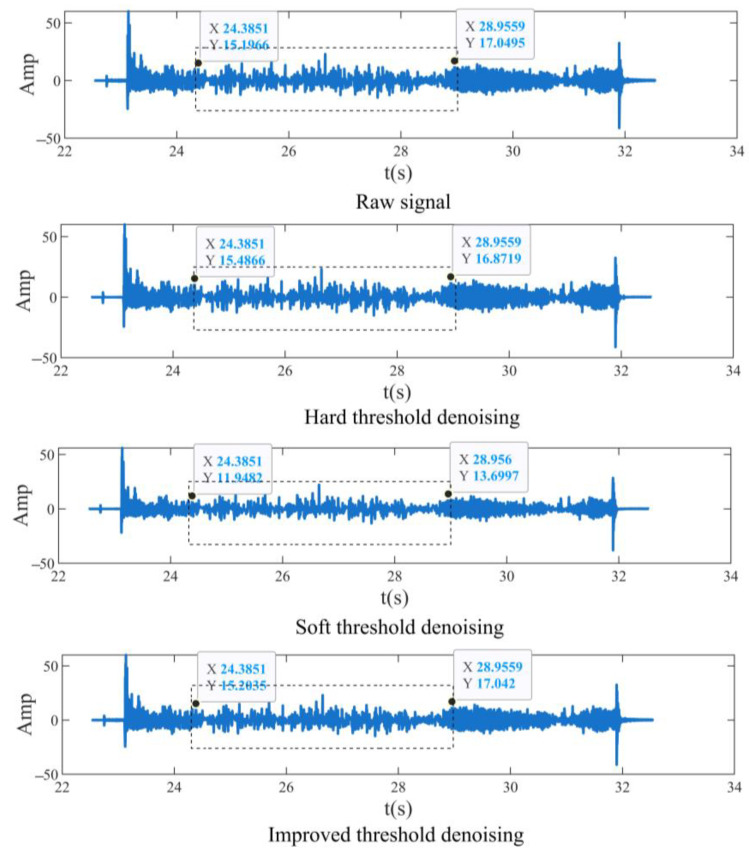
Comparison of denoising raw signals by different threshold functions.

**Figure 8 sensors-23-04790-f008:**
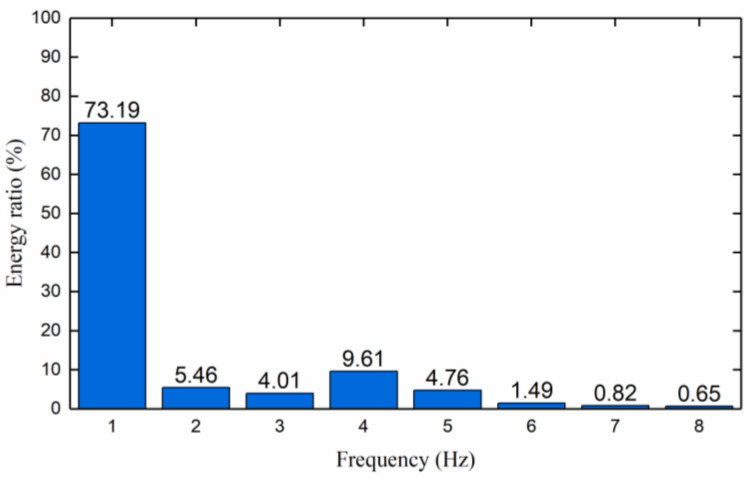
The energy ratio of each sub-band after NWTD-WP feature extraction of the signal.

**Figure 9 sensors-23-04790-f009:**
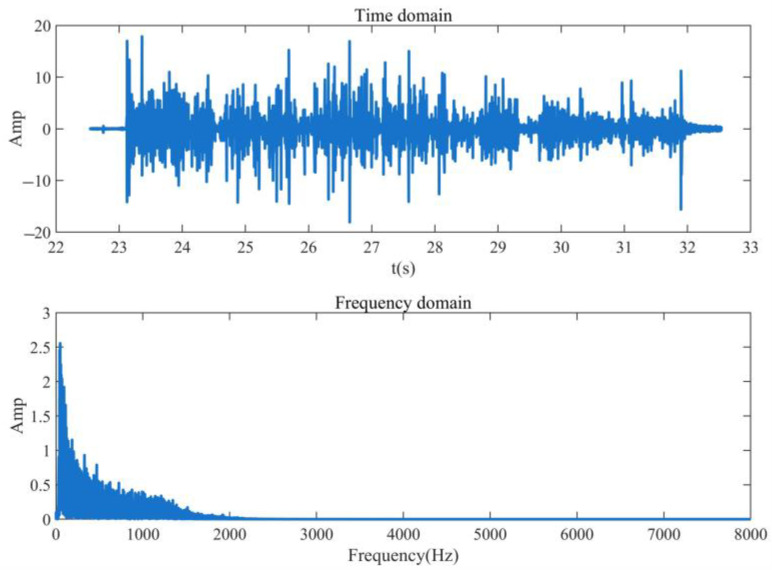
Time domain and frequency domain waveforms of NWTD-WP feature extraction for the first sub-band signal.

**Figure 10 sensors-23-04790-f010:**
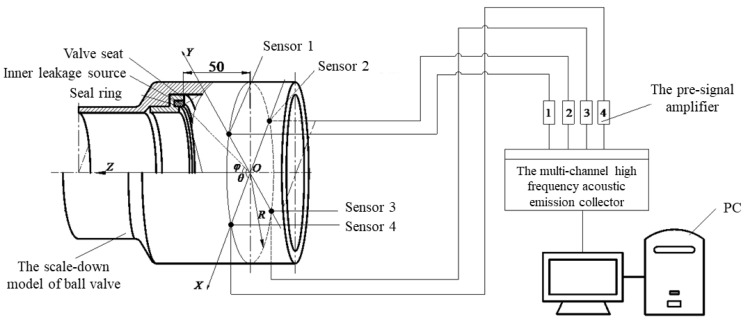
Schematic diagram of the internal leakage signal acquisition system for the scaled-down model of a large-diameter pipeline ball valve.

**Figure 11 sensors-23-04790-f011:**
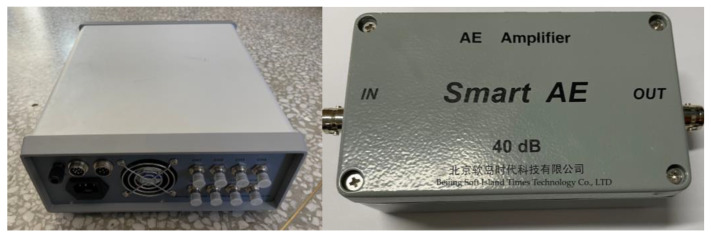
The multi-channel, high-frequency dynamic acoustic emission instrument and pre-signal amplifier.

**Figure 12 sensors-23-04790-f012:**
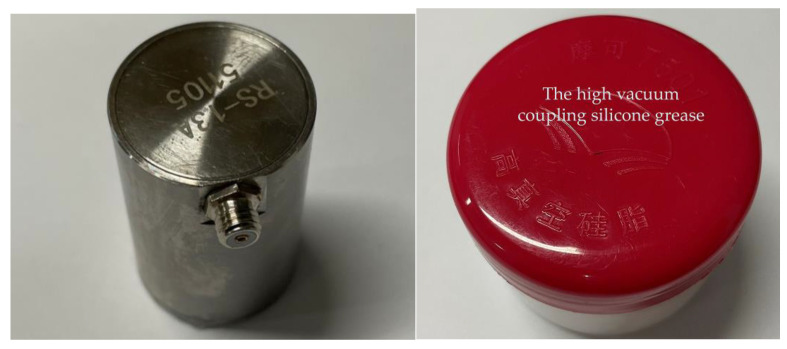
Acoustic emission sensor and the high vacuum coupling silicone grease.

**Figure 13 sensors-23-04790-f013:**
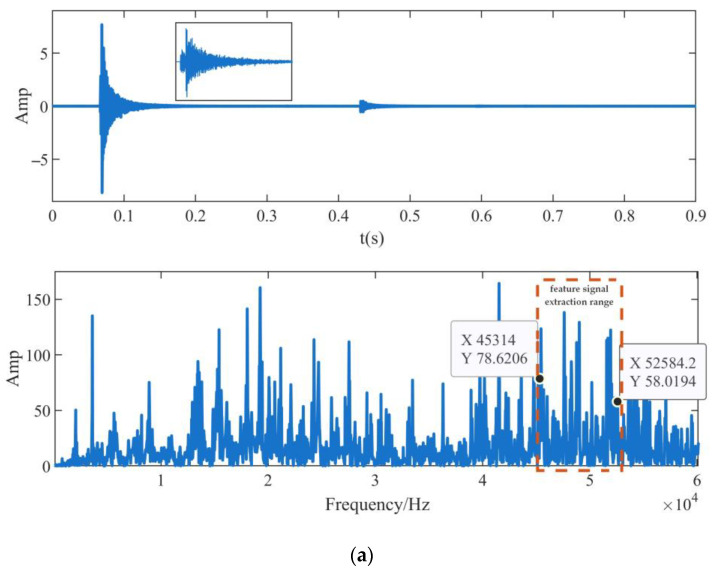
Time domain and frequency domain waveforms of internal leakage signal in the reduced-scale model of large-diameter pipeline ball valve. (**a**) sensor 1; (**b**) sensor 2; (**c**) sensor 3; (**d**) sensor 4.

**Figure 14 sensors-23-04790-f014:**
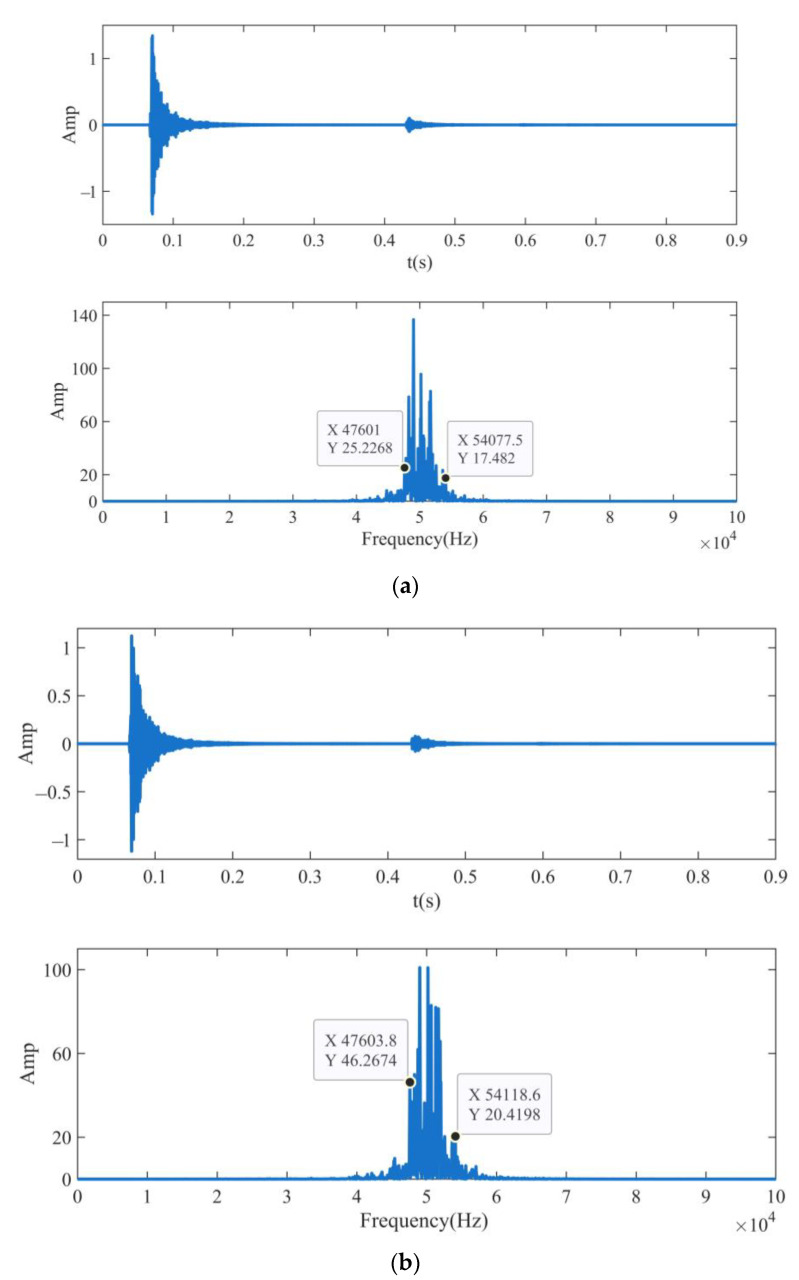
Time domain and frequency domain waveforms of NWTD-WP feature extraction of the seventh sub-band feature signal. (**a**) sensor 1; (**b**) sensor 2; (**c**) sensor 3; (**d**) sensor 4.

**Table 1 sensors-23-04790-t001:** The comparison of the noise reduction effect of different threshold quantization functions.

Threshold Function		Evaluation Index	
		SNR/dB	RMSE/dB
Soft Threshold		15.3933	0.4433
Hard Threshold		11.0999	0.7267
The improved threshold quantization function	*α* = 1, *β =* 0	14.8449	0.5754
	*α* = 0.03, *β =* 3	36.2026	0.0404
	*α* = 0.1, *β =* 10	34.5644	0.0488
	*α* = 0.25, *β =* 25	27.8835	0.1052
	*α* = 0.5, *β =* 50	23.6686	0.1710

**Table 2 sensors-23-04790-t002:** Parameters of high-frequency acoustic emission sensor.

Parameters of Sensor	Value
Model	RS-13A
The range of frequency	16~60 kHz
The center frequency	40 kHz
Detection surface (contact surface)	ceramics
Diameter	23.5 mm
Height	36 mm
The range of temperature	−20~130 °C
Interface	M5-KY

## Data Availability

Not applicable.

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
