# Peer review of "Research on Signal Feature Extraction of Natural Gas Pipeline Ball Valve Based on the NWTD-WP Algorithm"

_sensors, 2023, doi:10.3390/s23104790_

Round 1

Reviewer 1 Report

This paper presents an interesting approach  to the signal feature extraction of natural gas pipeline ball valve by using an improved wavelet threshold denoising. The effectiveness of the proposed method has been verified, combined with experiments. My suggestions include:

(1) In the introduction, the contributions of the paper should be highlighted. The shortcomings of existing methods should be analyzed and the advantages of the proposed  method should be made clear.

(2) The first point of the conclusive remarks should be explained in the introduction instead of conclusion section since WP is a common method in signal feature extraction. 

(3) The proposed method is expected to be compared with others existing methods in literature to verify its advantages over others. Moreover, its limitation should be explained in the conclusion.

The English writing should be improved. The authors are advised to go through the manuscript to correct grammar errors and typos. Specific abbreviations should be explained when they first appears, e.g. SNR, RC, EMD, VMD etc. 

Author Response

The reviewer's suggestion was accepted and the manuscript has been already modified . Please see the file named "review 1" for details.

Reviewer 2 Report

The paper presents a NWTD-WP algorithm for detecting valve leakage vibration. The paper is, well-written, original and useful, but there are some combined minor/major revisions that must be carried out in order to reconsider it for publication, which are: 

- State the abbreviation of NWTD-WP in the abstract (minor).

- The literature review is not well-written and there is no enough literature discussed (major).

- The paper contribution is not clear enough (major).

- mapping those signals (figure.12 to the practical issue of leakage or pipe break). The discussion part is sketchy (major).

- Support the conclusion by feasible future trends (minor). 

Best wishes to the authors

The English quality is fine. Minor proofreading is needed. 

Author Response

The reviewer's suggestion was accepted and the manuscript has been already modified . Please see the file named "review 2" for details.

Reviewer 3 Report

1. WP feature extraction and simulation is not clearly described in the manuscript.

2. Figure 2 Time domain waveforms of each sub-band after 3-layer wavelet decomposition of signal: please explain in a better way in the text.

3. What does WTD propose to eliminate the noise component in the signal y(t)?

4. Please read the instructions for authors and modify the structure of the manuscript.

5. Please improve the English written of the manuscript.

6. Figure 5 Threshold function curve comparison: this figure is ambiguous for readers. Please clarify it.

Please improve the English written of the manuscript.

Author Response

The reviewer's suggestion was accepted and the manuscript has been already modified . Please see the file named "review 3" for details.

Round 2

Reviewer 1 Report

The authors revise the paper carefully. My concerns have been well answered .

English writing has been improved.

Reviewer 2 Report

Thanks to the authors for the useful response. Recommending acceptance as is. 

English quality is fine. 

Reviewer 3 Report

Well-revised

Some minor modifications